# Pott Disease: A Tale of Two Cases

**DOI:** 10.3390/pathogens10091158

**Published:** 2021-09-09

**Authors:** Christopher Radcliffe, Matthew Grant

**Affiliations:** Section of Infectious Diseases, Department of Internal Medicine, Yale University School of Medicine, New Haven, CT 06510, USA; m.grant@yale.edu

**Keywords:** tuberculosis, osteomyelitis, Pott disease, HIV

## Abstract

Tuberculosis is considered one of the great masqueraders alongside syphilis and vasculitis. Pott disease is recognized as a classic manifestation of tuberculosis, yet it stands as a rare infectious syndrome in regions with low tuberculosis disease burden. To illustrate the challenges of diagnosing Pott disease in these settings, we report two cases and offer a brief overview of management recommendations for vertebral osteomyelitis caused by *Mycobacterium tuberculosis*. Case one concerns an 81-year-old man with a remote history of incarceration who presented with altered mental status and new pleural effusions. Case two is a 49-year-old man with well-controlled HIV who was transferred to our institution after being found to have extensive destruction of L3–L5 vertebrae and bilateral iliopsoas abscesses on outpatient imaging. These stand as illustrative examples of low and high suspicion for tuberculosis, respectively, and both cases required complex diagnostic and management decisions.

## 1. Introduction

The importance of tuberculosis cannot be overstated. Globally, millions of cases occur each year [1], and an estimated 1.7 billion people were living with latent tuberculosis infection (LTBI) in 2014 [2]. Tuberculosis disproportionately impacts vulnerable patient populations [3], case finding can prove challenging [1], and its range of clinical syndromes may hinder the diagnosis in regions with low disease burden. Following Percivall Pott’s 18th century treatise on spinal disease [4], tuberculosis of the spine is often referred to as Pott disease. A nationwide register-based study (*n* = 153) on Pott disease in Denmark reported a roughly three-week time lapse (median 19.5 days) between hospital contact and diagnosis, with 40% of patients requiring surgery and nearly 60% developing sequelae [5].

To highlight the clinical challenges of Pott disease, we report two cases treated in a low endemicity region. Case one concerns an 81-year-old man with a remote history of incarceration who presented with altered mental status and new pleural effusions. Case two is a 49-year-old man with well-controlled HIV who was transferred to our institution after being found to have extensive destruction of L3–L5 vertebrae and bilateral iliopsoas abscesses on outpatient imaging. These stand as illustrative examples of low and high suspicion for tuberculosis, respectively, and both cases required complex diagnostic and management decisions. We also present a brief review of management strategies for Pott disease.

## 2. Case Presentation

### 2.1. Case One

An 81-year-old man with diabetes mellitus, atrial fibrillation, and chronic lower back pain was admitted after being found unresponsive and hypoglycemic. After the return of euglycemia, the patient reported decreased appetite and a recent uptick in his insulin requirement. Physical exam was reportedly normal aside from mild confusion and an irregularly irregular rhythm. Laboratory studies were significant for mild hyponatremia (Na 130 mmol/L; reference range 135–145 mmol/L), anemia (11.3 g/dL; ref. range 12–18 g/dL), and lymphopenia (800/μL; ref. range 1000–4000/μL). A computed-tomography (CT) scan of the head without contrast was unremarkable, but a chest radiograph uncovered a moderately sized, right pleural effusion. Two sets of peripheral blood cultures were obtained.

The following day, additional studies revealed elevated pro B-type natriuretic peptide (2337 pg/mL; ref. range: <300 pg/mL), sedimentation rate (39 mm/hr; ref. range: 0–20 mm/hr), and high sensitivity C-reactive protein (hsCRP) (69 mg/L; ref. range: 0.1–3 mg/L). A CT scan of the chest without contrast demonstrated findings compatible with discitis and osteomyelitis at T10–T11 level (Figure 1), moderate to large bilateral pleural effusions with associated compressive atelectasis of the adjacent lower lobes, consolidation of the right middle lobe, cardiomegaly, a small pericardial effusion, lung nodules measuring up to 1 cm, scattered calcified granulomas, and mildly enlarged intrathoracic lymph nodes.

A transthoracic echocardiogram showed progression of aortic stenosis, mild mitral and aortic regurgitation, and preserved ejection fraction (50–55%). Follow-up magnetic resonance imaging (MRI) with contrast confirmed discitis–osteomyelitis at T10–T11 with epidural phlegmon or abscess causing severe central canal stenosis. Further history revealed that he had been experiencing dyspnea on exertion for roughly three months.

A right thoracentesis was performed on hospital day (HD) 2 with a return of 2 L of bloody pleural fluid. Pleural fluid studies were consistent with an exudative effusion and demonstrated 19,000 red cells/μL along with 2857 nucleated cells/μL (64% lymphocytes). Cytology was negative for malignant cells but noted chronic inflammation. Bacterial, fungal, and acid-fast bacillus (AFB) cultures were also sent and ultimately returned negative. Routine sputum culture on HD3 showed 2+ normal flora. Peripheral blood cultures collected on admission were reported as negative on HD5.

Given the continued concern for malignancy, a repeat right thoracentesis was performed on HD9. Pleural fluid studies were again exudative with 9000 red cells/μL, 2575 nucleated cells/μL (77% lymphocytes), and negative adenosine deaminase (6.2 U/L). Repeat cytology studies were unchanged. Bacterial culture of pleural fluid was sent and returned negative. Infectious diseases was consulted the same day, and further history revealed subacute cough with minimal sputum and an 18 kg weight loss in the preceding six months. The patient also reported a remote history of brief incarceration. He was transferred to a negative-pressure room given concern for active tuberculosis.

Three AFB sputum cultures were collected. On HD13, QuantiFERON^®^-TB Gold resulted as positive (0.83 IU/mL; negative <0.35 IU/mL). A CT-guided biopsy of the left inferior T10 endplate was obtained on HD16, and pathology returned as consistent with discitis/osteomyelitis. AFB staining of the biopsy specimen was negative, and the specimen was only sent to pathology due to inadequate size for both pathology and culture. Continued diagnostic uncertainty and persistent effusion prompted a thoracoscopy with right pleural biopsy, chest tube placement, and tunneled pleural catheter placement on HD22. Nocardia, actinomyces, AFB, bacterial, and fungal tissue cultures were sent in addition to repeat AFB, bacterial, and fungal cultures of pleural fluid. All eventually returned negative. A DNA probe for *Mycobacterium tuberculosis* (MTB) DNA was sent on pleural fluid and returned negative.

One AFB sputum culture was reported as growing *Mycobacterium avium* complex (MAC) on HD26, and the pleural biopsy pathology resulted as marked granulomatous inflammation with negative AFB staining. HIV testing was negative. Treatment for MAC (clarithromycin, ethambutol, rifabutin) was initiated on HD28 then held due to continued uncertainty with regards to etiology of osteomyelitis. The pleural biopsy specimen was sent to the University of Washington for broad-range mycobacterial polymerase chain reaction (PCR); negative results were obtained after several weeks. After discussion with the patient, empiric therapy (rifabutin, isoniazid, pyrazinamide, ethambutol) for MTB was initiated on HD36. Before discharge on HD40, azithromycin was added to cover possible MAC.

The day after discharge, an AFB sputum culture from HD11 was reported as growing MTB complex. This finding was later confirmed by the Connecticut Department of Public Health (CT DPH). The patient was readmitted given the need for negative-pressure isolation then discharged after continuing antimycobacterial therapy for 12 days and providing two sputum specimens with negative AFB smears. Azithromycin was discontinued upon discharge. Weeks after discharge, *Mycobacterium gordonae* grew from one of two AFB sputum cultures and was deemed to be non-pathogenic. Roughly one month after discharge, he was seen in clinic and reported interval improvement in back pain. A plan for 12 months of therapy was made, with pyrazinamide and ethambutol being discontinued after the initial two months. Unfortunately, six months into therapy, he died after being admitted for altered mental status, failure to thrive, and multidrug-resistant (MDR) *Citrobacter freundii* urinary tract infection.

### 2.2. Case Two

A 49-year-old man with well-controlled HIV (CD4 count 625/μL and undetectable viral load on elvitegravir, cobicistat, emtricitabine, and tenofovir alafenamide) was transferred to our institution due to concern for Pott disease. Eight months before transfer, he had a positive purified protein derivative test in the context of intermittent hemoptysis, and a chest radiogram was reportedly normal. Five months before transfer, isoniazid was initiated as treatment for LTBI. Two months before transfer, he was admitted at an outside hospital for hemoptysis, and an AFB sputum culture reportedly grew MAC.

Two weeks before transfer, he was sent to the emergency department of an outside hospital after an outpatient MRI showed marked destruction of L3–L5 vertebral bodies with gibbus deformity, diffuse ventral epidural phlegmon from L3–L5, and diffuse paraspinal and psoas abscesses. When evaluated at the outside hospital, he reported several months of worsening back pain, subjective fever, night sweats, and difficulty with ambulation for several weeks. Notably, he had chronic back pain following a motorcycle accident six years prior. He was born in the Caribbean and had immigrated to the United States less than a year before evaluation. He had been living with HIV for roughly one decade and reported being treated for pulmonary tuberculosis as a child.

During admission at the outside hospital, a CT-guided vertebral biopsy of the lumbar region was non-diagnostic with a negative AFB smear. Empiric piperacillin–tazobactam was initiated then discontinued. Aspiration of left paraspinal musculature was AFB smear-negative. Both the biopsy and aspirate AFB cultures were negative at the time of transfer to our institution. Karius (Karius, Redwood City, CA, USA) cell-free DNA testing reportedly detected a low level of MTB complex DNA in the aspirate sample, and RIPE (rifampin, isoniazid, pyrazinamide, ethambutol) was initiated. HIV medication was transitioned to emtricitabine–tenofovir and dolutegravir to avoid drug–drug interaction between cobicistat–rifampin co-administration. At the time of transfer, piperacillin–tazobactam had been reinitiated due to fever (T_max_ 39.2 °C).

On admission to our institution, RIPE was continued and other antibiotics discontinued. Frank gibbus deformity and tenderness in the lumbar region were noted on exam. Laboratory studies showed leukocytosis (14,500/μL; ref. range: 4000–10,000/μL), anemia (9.2 g/dL), thrombocytosis (621 × 10^3^/μL; ref. range: 140–440 × 10^3^/μL), elevated hsCRP (178.6 mg/L), and elevated sedimentation rate (117 mm/hr). A repeat CT scan of the lumbar spine without contrast on HD2 re-demonstrated prior findings (Figure 2).

Three AFB sputum cultures were collected and eventually returned negative. Moxifloxacin was added to the treatment regimen on HD3 given concern for resistance due to recent LTBI treatment. A CT scan of the chest, abdomen, and pelvis was performed and showed bilateral iliopsoas abscesses tracking to bilateral inguinal regions.

A CT-guided biopsy of L4 vertebral body and left paraspinal abscess aspiration were performed on HD7. Specimens were sent for AFB staining, culture, pathology, and real-time PCR sequencing. Pathology of vertebral biopsy showed mild chronic osteomyelitis with negative AFB staining and no granulomatous inflammation. The left paraspinal abscess aspirate cultures ultimately grew MTB complex and coagulase-negative Staphylococcus in the following weeks, with the latter being deemed a contaminant. MTB complex DNA in the aspirate was also detected via real-time PCR. Bilateral inguinal collections were drained on HD9 by interventional radiology with the placement of three drainage catheters. Three separate specimens were sent for AFB culture, and one of three was smear-positive with 1+ AFB the following day. The specimen was sent to CT DPH for further testing. In subsequent weeks, all three cultures grew MTB complex.

Scrotal pain developed on HD13 and an ultrasound confirmed a 10.4 × 5.4 × 9.8 cm hydrocele. Urology was consulted and recommended against aspiration given the likely chronic nature of the hydrocele. An AFB urine culture was collected on HD15 and ultimately returned negative. On HD21, CT DPH reported a rifampin resistance gene per Gene Xpert (Cepheid, Sunnyvale, CA, USA) testing, and amikacin was added given concern for MDR MTB. Further genotypic evidence of resistance was noted for isoniazid, rifampin, and pyrazinamide on HD24, so these drugs were discontinued. The following day, linezolid and ethionamide were added to the treatment regimen. Ethionamide was held due to severe nausea and vomiting, and abdominal pain led to a CT scan of the abdomen and pelvis which revealed new destructive changes of the L2 vertebral body.

New diplopia and headache on HD27 prompted consideration of tuberculous meningitis. The following day, a CT scan of the head without contrast was ordered after an unresponsive episode, but returned unremarkable. A follow-up MRI brain with contrast showed mild, diffuse thickening of meninges deemed to be non-specific. Cycloserine was added, and a steroid taper was initiated for possible paradoxical reaction to treatment. Lumbar puncture was performed on HD29, and cerebrospinal fluid (CSF) studies were unremarkable. Bacterial and AFB cultures of CSF ultimately returned negative as did MTB PCR of CSF. Bilateral inguinal collections were again drained by interventional radiology on HD35, but purulent fluid was only sent for routine bacterial culture which returned negative.

Ethionamide was restarted without event on HD39. For ease of administration, HIV medication was changed to bictegravir–emtricitabine–tenofovir alafenamide on HD49. Phenotypic susceptibility testing was received from the Centers for Disease Control and Prevention on HD55 and confirmed resistance to rifampin, isoniazid, and pyrazinamide. After approval of request by CT DPH, bedaquiline was added to the regimen on HD68. In the following week, cycloserine and ethionamide were discontinued. The drug regimen was transitioned to amikacin, bedaquiline, ethambutol, linezolid, and moxifloxacin. He was ultimately discharged on HD87.

One week after discharge, he was seen in clinic and reported improvement in lower extremity strength. Shortly thereafter, he transferred his tuberculosis care to an outside institution. Notably, six months after discharge, he was evaluated by neuro-ophthalmology at our institution for drug-induced optic neuropathy possibly attributable to ethambutol, linezolid, or ethionamide.

## 3. Discussion

We report two cases of Pott disease from a region with low disease burden. In this setting, knowledge of the epidemiology of tuberculosis is vital. Incidence rates certainly vary across countries, and the United States has a low tuberculosis rate reported as 2.7 cases per 100,000 persons in 2019 [6]. Roughly 70% of cases from 2019 impacted persons born outside of the United States, and the majority of cases represented reactivation of LTBI [6]. It is interesting to note that tuberculosis cases in the United States between 2014 and 2017 were diagnosed in older individuals relative to cases in 2010–2013 [7], suggesting that the index of suspicion may need to be heightened when evaluating older adults. Finally, it is prudent to consider HIV co-infection, prior history of incarceration, or housing instability when evaluating for tuberculosis in low endemicity regions [3].

Consistent with prior reports [8], both of our cases experienced several months of symptoms before being diagnosed. Low suspicion for tuberculosis characterized the initial course for case one whereas high suspicion for tuberculosis was continually present in the latter case. Nonetheless, several diagnostic maneuvers were required in both cases before a definitive clinical picture emerged. Disseminated disease was evident, and management rightfully differed based on comorbidities, susceptibility results, and severity of disease. Neither case underwent neurosurgical intervention due to perceived risks associated with age and the extent of disease in cases one and two, respectively.

In general, spinal tuberculosis is distinct from bacterial osteomyelitis, and the pathogenesis is thought to involve paravertebral venous spread as opposed to arterial hematogenous seeding of bone [9]. Treatment is challenging due to the need for bone penetration with antimicrobials coupled by the paucibacillary nature of spinal tuberculosis, as opposed to the multibacillary form in highly oxygenated lung parenchyma [9]. Furthermore, most high-quality studies have traditionally focused on pulmonary tuberculosis [9,10]. For spinal tuberculosis, Infectious Diseases Society of America (IDSA) guidelines recommend 6–9 months of therapy for drug-susceptible MTB, with experts recommending 12-month courses for patients with spinal hardware [10]. Treatment of MDR tuberculosis is more complex [11], and current IDSA guidelines are largely informed by a recent meta-analysis showing mortality benefits with linezolid, fluoroquinolones, and bedaquiline [11,12]. The role of surgical intervention in Pott disease has been controversial due to the heterogeneity of this syndrome [9], but a study on United States data from 2002 to 2011 reported 22% of patients having undergone surgery, with 50% of these cases requiring intervention at multiple spinal levels [13]. Prevalence of surgical or procedural intervention has ranged from 10.4% to 67% in other recent cohorts [5,14,15]. Outcomes vary widely [5,8,9,14,15], and lasting neurological sequalae are not uncommon.

## 4. Conclusions

Pott disease is a challenging, heterogeneous condition. Both of our cases experienced diagnostic delays and underwent multiple procedural interventions with diagnostic and therapeutic intent. Active tuberculosis continues to impact millions [1], and attention to the individual patient is vital in the treatment of extrapulmonary manifestations.

## Figures and Tables

**Figure 1 pathogens-10-01158-f001:**
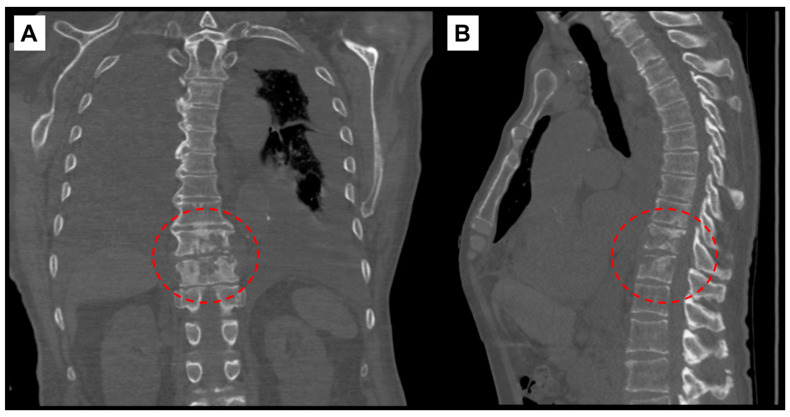
Computed-tomography scan of chest showing T10–T11 discitis and osteomyelitis. (**A**) Coronal view with red, interrupted circle highlighting destructive changes to vertebral bodies. (**B**) Sagittal view.

**Figure 2 pathogens-10-01158-f002:**
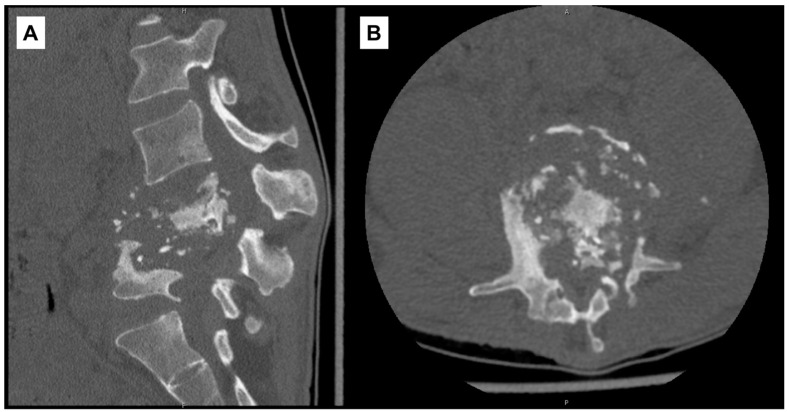
Computed-tomography scan of lumbar spine with diffuse destruction of L4–L5 vertebral bodies. (**A**) Sagittal view. (**B**) Axial view at L4 vertebral level.

## Data Availability

Not applicable.

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
