# Peer review of "Pott Disease: A Tale of Two Cases"

_pathogens, 2021, doi:10.3390/pathogens10091158_

Round 1

Reviewer 1 Report

Two complex cases that required extensive work up  with challenging decision making . Cases are well presented.

This is a well-written and interesting case series (2 cases of spinal/paravertebral tuberculosis).  It provides an interesting discussion of the clinical approach to diagnosis.   

Author Response

Thank you for your comments.

Reviewer 2 Report

This case report describes about clinical course, diagnostic procedures, and treatment trials of two Pott disease cases. I think that it provides invaluable information for the clinician in this field. Just one thing, although authors have described some hint about infection route or sources, it should be need more details discussion about this issue in lower TB endemic country.   

Author Response

Thank you for your comments. In order to strengthen our discussion, we have added the following comments to lines 202-211:

In this setting, knowledge of the epidemiology of tuberculosis is vital. Incidence rates certainly vary across countries, and the United States has a low tuberculosis rate reported as 2.7 cases per 100,000 persons in 2019 [6]. Roughly 70% of cases from 2019 impacted persons born outside of the United States, and the majority of cases represented reactivation of LTBI [6]. It is interesting to note that tuberculosis cases in the United States between 2014-2017 were diagnosed in older individuals relative to cases in 2010-2013 [7], suggesting that the index of suspicion may need to be heightened when evaluating older adults. Finally, it is prudent to consider HIV co-infection, prior history of incarceration, or housing instability when evaluating for tuberculosis in low endemicity regions [3].
